# Surveillance, Isolation, and Genetic Characterization of Bat Herpesviruses in Zambia

**DOI:** 10.3390/v15061369

**Published:** 2023-06-13

**Authors:** Hayato Harima, Yongjin Qiu, Junya Yamagishi, Masahiro Kajihara, Katendi Changula, Kosuke Okuya, Mao Isono, Tomoyuki Yamaguchi, Hirohito Ogawa, Naganori Nao, Michihito Sasaki, Edgar Simulundu, Aaron S. Mweene, Hirofumi Sawa, Kanako Ishihara, Bernard M. Hang’ombe, Ayato Takada

**Affiliations:** 1Laboratory of Veterinary Public Health, Faculty of Agriculture, Tokyo University of Agriculture and Technology, Tokyo 183-8509, Japan; harima@go.tuat.ac.jp (H.H.); kanako-i@cc.tuat.ac.jp (K.I.); 2Management Department of Biosafety, Laboratory Animal and Pathogen Bank, National Institute of Infectious Diseases, Tokyo 162-8640, Japan; qiu-y@niid.go.jp; 3Department of Virology-I, National Institute of Infectious Diseases, Tokyo 162-8640, Japan; 4Division of Collaboration and Education, International Institute for Zoonosis Control, Hokkaido University, Sapporo 001-0020, Japan; junya@czc.hokudai.ac.jp; 5International Collaboration Unit, International Institute for Zoonosis Control, Hokkaido University, Sapporo 001-0020, Japan; 6Division of International Research Promotion, International Institute for Zoonosis Control, Hokkaido University, Sapporo 001-0020, Japan; kajihara@czc.hokudai.ac.jp (M.K.); n-nao@czc.hokudai.ac.jp (N.N.); 7Department of Para-Clinical Studies, School of Veterinary Medicine, The University of Zambia, P.O. Box 32379, Lusaka 10101, Zambia; katendi.changula@sacids.org (K.C.); mudenda68@yahoo.com (B.M.H.); 8Department of Pathogenetic and Preventive Veterinary Science, Joint Faculty of Veterinary Medicine, Kagoshima University, Kagoshima 890-0065, Japan; kokuya@vet.kagoshima-u.ac.jp; 9Division of Global Epidemiology, International Institute for Zoonosis Control, Hokkaido University, Sapporo 001-0020, Japan; maoisono@gmail.com; 10Department of Oral Microbiology, Faculty of Medicine, Dentistry and Pharmaceutical Sciences, Okayama University, Okayama 700-8558, Japan; tomoyuki.yamaguchi@okayama-u.ac.jp; 11Department of Virology, Faculty of Medicine, Dentistry and Pharmaceutical Sciences, Okayama University, Okayama 700-8558, Japan; hogawa@okayama-u.ac.jp; 12One Health Research Center, Hokkaido University, Sapporo 060-0818, Japan; h-sawa@ivred.hokudai.ac.jp; 13Division of Molecular Pathobiology, International Institute for Zoonosis Control, Hokkaido University, Sapporo 001-0020, Japan; m-sasaki@czc.hokudai.ac.jp; 14Department of Disease Control, School of Veterinary Medicine, The University of Zambia, P.O. Box 32379, Lusaka 10101, Zambia; esikabala@yahoo.com; 15Macha Research Trust, Choma 20100, Zambia; 16Africa Center of Excellence for Infectious Diseases of Humans and Animals, The University of Zambia, P.O. Box 32379, Lusaka 10101, Zambia; 17Institute for Vaccine Research and Development, Hokkaido University, Sapporo 001-0021, Japan

**Keywords:** herpesvirus, bat, surveillance, complete genome, Zambia

## Abstract

Bats are of significant interest as reservoirs for various zoonotic viruses with high diversity. During the past two decades, many herpesviruses have been identified in various bats worldwide by genetic approaches, whereas there have been few reports on the isolation of infectious herpesviruses. Herein, we report the prevalence of herpesvirus infection of bats captured in Zambia and genetic characterization of novel gammaherpesviruses isolated from striped leaf-nosed bats *(Macronycteris vittatus*). By our PCR screening, herpesvirus DNA polymerase (DPOL) genes were detected in 29.2% (7/24) of Egyptian fruit bats (*Rousettus aegyptiacus*), 78.1% (82/105) of *Macronycteris vittatus*, and one Sundevall’s roundleaf bat (*Hipposideros caffer*) in Zambia. Phylogenetic analyses of the detected partial DPOL genes revealed that the Zambian bat herpesviruses were divided into seven betaherpesvirus groups and five gammaherpesvirus groups. Two infectious strains of a novel gammaherpesvirus, tentatively named Macronycteris gammaherpesvirus 1 (MaGHV1), were successfully isolated from *Macronycteris vittatus* bats, and their complete genomes were sequenced. The genome of MaGHV1 encoded 79 open reading frames, and phylogenic analyses of the DNA polymerase and glycoprotein B demonstrated that MaGHV1 formed an independent lineage sharing a common origin with other bat-derived gammaherpesviruses. Our findings provide new information regarding the genetic diversity of herpesviruses maintained in African bats.

## 1. Introduction

Bats are associated with several emerging zoonotic viruses such as Marburg virus, Hendra virus, and Nipah virus, and are thought to act as natural reservoirs for these emerging viruses [1]. Global viral surveillance aiming at identifying and mitigating future emergence of bat-borne zoonotic outbreaks has identified 12,986 bat-associated viruses as of August 2021 [2]. Bats (order *Chiroptera*) are the second most diverse mammalian order, comprising more than 20% of all known mammalian species [3]. The high viral diversity in bats relative to other mammals is reflective of the number of species [4]. Novel viral sequences are constantly discovered in various bat species owing to the development of next-generation sequencing technologies (NGS) [1].

Herpesviruses are enveloped, double-stranded DNA (dsDNA) viruses belonging to the family *Herpesviridae* that are classified into three subfamilies (*Alphaherpesvirinae*, *Betaherpesvirinae*, and *Gammaherpesvirinae*) on the basis of their genomic architectures, sequence similarities, and biological properties [5]. The dsDNA genome size of herpesviruses is approximately 120–240 kbp, including approximately 70–170 open reading frames (ORFs), and 40 of which encode core viral proteins conserved among all known herpesviruses [6,7]. The core viral proteins are generally involved in fundamental aspects of the viral life cycle such as nucleic acid metabolism, DNA synthesis, DNA packaging, and virion maturation, and are consequently often essential for viral replication [5]. In addition to ORFs, most herpesvirus genomes contain various reiterated sequences such as terminal, internal, and/or inverted repeats [5]. To date, 17 genera and 118 species have been classified into the *Herpesviridae* family by the International Committee on Taxonomy of Viruses (ICTV) [6]. In addition to these classified herpesviruses, many related and unclassified family members have been identified in various hosts, and the number of herpesvirus species is likely to exceed more than 200 [5]. In general, the host ranges of individual herpesviruses are restricted with most having evolved in association with their own host species [6].

Alpha-, beta-, and gammaherpesviruses have been discovered in various bat species worldwide, showing the high diversity of herpesviruses circulating in the hosts [8]. According to the database of zoonotic and vector-borne viruses (ZOVER), 348 complete or partial herpesvirus genomic sequences have been identified in 75 bat species as of 2022 [2]. In the last two decades, surveillance of herpesvirus infection in bats has been reported in African countries, including Cameroon, the Central African Republic, Ghana, Kenya, Madagascar, the Republic of the Congo, and South Africa [9,10,11,12,13]. In previous studies, herpesviruses were identified in a total of nine bat species in Africa: three species in the family *Pteropodidae* (frugivorous bats; *Eidolon dupreanum*, *Eidolon helvum*, and *Rousettus aegyptiacus*), four species in the family *Vespertilionidae* (insectivorous bats; *Miniopterus natalensis*, *Neoromicia capensis*, *Neoromicia helios*, and *Pipistrellus nanulus*), and two species in the family *Rhinolophidae* (insectivorous bats; *Triaenops afer* and *Triaenops persicus*). Although numerous partial herpesvirus genes have been detected by genetic screening, most of the detected herpesviruses remain to be isolated. To date, eleven herpesviruses have been isolated from bats. Six alphaherpesviruses were isolated from frugivorous bats (*Eidolon dupreanum*, *Eidolon helvum*, *Pteropus lylei*, and *Pteropus hypomelanus*-related bats), insectivorous bats (*Lonchophylla thomasi*), and an unidentified bat [13,14,15]; two betaherpesviruses were isolated from insectivorous bats (*Miniopterus fuliginosus* and *Miniopterus schreibersii*) [16,17], and three gammaherpesviruses were isolated from insectivorous bats (*Myotis velifer incautus*, *Eptesicus fuscus*, and *Rhinolophus ferrumequinum*) [18,19,20]. Of these, six isolates (two alphaherpesviruses, one betaherpesvirus, and three gammaherpesviruses) were analyzed to determine their complete or nearly complete genome sequences. The bat herpesviruses for which complete genome sequences are available have been identified for bats in Oceania (Australia), Asia (Vietnam, Indonesia, and Japan), and North America (United States of America and Canada), but no complete genome sequence of herpesviruses detected in African bats has been reported.

As part of the surveillance program of zoonotic virus infection in bats in Zambia, we screened frugivorous bats (Egyptian fruit bats, *Rousettus aegyptiacus*) and insectivorous bats (striped leaf-nosed bats, *Macronycteris vittatus*; Sundevall’s roundleaf bats, *Hipposideros caffer*) for herpesviruses, followed by phylogenetic analysis of the detected sequences. Here, we also report the isolation of a novel gammaherpesvirus named Macronycteris gammaherpesvirus 1 (MaGHV1) from *Macronycteris vittatus* bats in Zambia. We performed complete genome sequencing and genetically characterized the isolates.

## 2. Materials and Methods

### 2.1. Sample Collection

In 2018, 24 cave-dwelling Egyptian fruit bats (*Rousettus aegyptiacus*), 105 striped leaf-nosed bats (*Macronycteris vittatus*), and one Sundevall’s roundleaf bat (*Hipposideros caffer*) were captured in Chongwe (15.6° S, 28.7° E) with approval from the Department of National Parks and Wildlife, Ministry of Tourism and Arts, Zambia (DNPW8/27/1) [21]. Bat species were identified based on morphological characteristics and nucleotide sequence analysis of the mitochondrial 16S ribosomal RNA and the cytochrome b genes. We collected oral and rectal swabs from each captured bat and promptly placed them in Eagle’s minimum essential medium (MEM) supplemented with 1000 units/mL penicillin, 1000 µg/mL streptomycin, 25 µg/mL amphotericin B, 0.01 M HEPES, and 0.5% bovine serum albumin (BSA). After brief centrifugation, the supernatants were stored at −80 °C.

### 2.2. Amplification of the Herpesvirus DNA Polymerase Gene by PCR

Screening for the herpesvirus detection was conducted using pooled swab specimens. Oral and rectal swabs were pooled for each bat and nucleic acids were individually extracted from 130 pooled swab specimens using the QIAamp Viral RNA Mini Kit (QIAGEN, Hilden, Germany) according to the manufacturer’s instructions. Nucleic acids extracted by the kit include not only RNA but also DNA as previously described [22]. The herpesvirus DNA genome was detected by semi-nested PCR using Tks Gflex DNA Polymerase (TaKaRa, Shiga, Japan) with pan-herpesvirus primer sets targeting the DNA polymerase (DPOL) gene (Appendix A) [23]. PCR products were subjected to direct sequencing using the Big Dye Terminator v3.1 Cycle Sequencing kit (Applied Biosystems, Foster City, CA). The 5′- and 3′-ends of the sequences derived from primers were trimmed and obtained sequences were analyzed through a BLAST search (https://blast.ncbi.nlm.nih.gov/Blast.cgi, accessed on 23 May 2023). The determined partial DPOL nucleotide sequences were deposited into the DNA Data Bank of Japan (DDBJ) under accession numbers LC762209-LC762247.

### 2.3. Virus Isolation

African green monkey kidney (Vero E6) cells were maintained in Dulbecco’s Modified Eagle’s Medium (DMEM) supplemented with 10% fetal bovine serum (FBS), 2 mM L-glutamine, 100 units/mL penicillin, 100 µg/mL streptomycin, 3.5 mg/mL D-glucose, and 1.0 mg/mL NaHCO_3_ at 37 °C with 5% CO_2_. The media from the swabs that tested positive for herpesvirus detection by genetic screening were inoculated onto Vero E6 cell cultures, followed by 1 h incubation at 37 °C in 5% CO_2_ for virus adsorption. After the inocula were removed, the cells were washed twice with PBS and maintained in DMEM containing 5 µg/mL trypsin, 0.3% BSA, 2 mM L-glutamine, 4% antibiotic–antimycotic solution (Gibco, Waltham, MA, USA), and 1.0 mg/mL NaHCO_3_ at 37 °C in 5% CO_2_ for 2 weeks. The supernatant of the inoculated cells was blindly passaged to fresh Vero E6 cells. Subsequently, the supernatant of the passaged culture from oral and rectal swabs was pooled for each bat and subjected to DNA extraction and NGS analysis as described below. Total DNA was extracted from the pool by using a DNeasy Blood & Tissue Kit (QIAGEN), and isolation of herpesviruses was confirmed by PCR.

### 2.4. Genome Sequencing, Construction, and Annotation

The complete genome sequences of isolated herpesviruses were determined by NGS analyses. The sequencing library was synthesized from total DNA of isolated viruses using a Nextera XT DNA Library Preparation Kit (Illumina, San Diego, CA, USA) according to the manufacturer’s instructions, and was then sequenced on a MiSeq instrument with a MiSeq Reagent Kit v3 (600 cycles) to generate 300 bp paired-end reads. Sequence reads were trimmed and analyzed by de novo assembly using CLC Genomics Workbench software (CLC bio, Hilden, Germany). Consensus sequences with coverage of over 20 reads were obtained from the herpesvirus genome contigs.

For filling the remaining gap, PCR was performed using a specific primer designed based on the upstream and downstream sequences of the gaps (Appendix A). The obtained amplicons were sequenced using a Frongle and Ligation Sequencing Kit (SQK-Q20EA) (Oxford Nanopore Technologies, Oxford, United Kingdom). Base calling was performed using Guppy version 5.0.16+b9fcd7b. Extra sequences were trimmed using Porechop [24]. Then, consensus sequences were generated using Canu [25]. To compliment the terminal repeat sequences at both ends of the genome, the genome DNA was amplified using an illustra GenomiPhi V2 DNA Amplification Kit (GE Healthcare, Chicago, IL, USA), and their terminals were repaired using T7 Endonuclease I (New England Biolabs, Ipswich, MA, USA). Following sequencing, base calling and trimming were performed as described above. To select reads including the terminal sequence, they were aligned with one kb regions flanking each terminal repeat using BWA [26]. The selected reads for each terminal were assembled using Canu, and the most representative contig was manually selected. The obtained gap-filling and terminal repeat sequences were concatenated with the genome assembly and polished with Pilon [27]. The determined genome sequences were deposited in the DDBJ database under accession numbers LC763829-LC763830.

ORFs were predicted by the ORF finder tool provided by the National Center for Biotechnology Information (NCBI) (https://www.ncbi.nlm.nih.gov/orffinder/, accessed on 23 May 2023). ORFs were identified using the following criteria: (i) sequence similarity to known viral or cellular genes, (ii) the presence of an initiating methionine and a stop codon, and/or (iii) length greater than 100 amino acids. ORFs with more than two of these criteria were considered to be genes encoding viral proteins. The translated amino acid sequences of the ORFs were subjected to BLAST search to determine whether they matched known viral or cellular proteins. The nomenclature used for MaGHV1 ORFs is based on a similar strategy used for other gammaherpesviruses such as equid herpesvirus 2 [28]. The unified name of the ORF for viral genes conserved across the subfamily *Gammaherpesvirinae* is used in this study. Additional ORFs are numbered in order of the appearance in the genome from 5′ to 3′. The complete sequence of the isolated virus was analyzed in a tandem repeat finder program [29] to identify the repeat regions.

### 2.5. Genetic Comparison and Phylogenetic Analysis

Bioinformatic analyses were performed using herpesvirus sequences deposited in the DDBJ/EMBL-Bank/GenBank databases. Identity comparison analyses were conducted between isolated herpesviruses using GENETYX version 15 (GENETYX Corporation, Tokyo, Japan). Phylogenetic analyses based on the amino acid sequences of DPOL and glycoprotein B (gB) genes were performed using MEGA X software [30]. The MUSCLE protocol was used to align the sequences. Phylogenetic trees were constructed using the maximum likelihood method based on the model of LG+G+I for complete and partial DPOL genes and the model of LG+G+I+F for the gB gene as the best-fit models, with 1000 bootstrap replicates. The viruses and their accession numbers of the sequences used in the analysis are listed in Appendix A.

## 3. Results

### 3.1. Screening for Herpesviruses with DPOL Gene by Semi-Nested PCR

Swab specimens collected from 130 bats captured in Zambia were screened for herpesviruses by PCR. Of these, 69.2% (90/130) were positive for herpesvirus detection (Table 1), suggesting high prevalence of herpesvirus infection in bats in Zambia; 29.2% (7/24) of *Rousettus aegyptiacus*, 78.1% (82/105) of *Macronycteris vittatus*, and one *Hipposideros caffer*. BLAST analyses revealed that 26 of the 90 obtained sequences (181 or 184 bp in length) originated from betaherpesviruses and that 60 of the sequences (172 bp in length) originated from gammaherpesviruses. These sequences showed 45.6–100% and 68.4–100% amino acid identities with known betaherpesviruses and gammaherpesviruses, respectively (Appendix A). Exceptionally, 4 sequences (172 bp in length of sample ID 3, 6, 114, and 126) were related to various alpha-, beta-, or gammaherpesviruses with 40–60% amino acid identities through BLAST analyses. Since these four sequences showed the highest identities with a bat-derived betaherpesvirus (*Plecotus austriacus*/Spain/psc-23/2004or2007), we tentatively classified them into the subfamily *Betaherpesvirinae* (Table 2).

### 3.2. Phylogenetic Analyses Based on the Partial DPOL Gene

Phylogenetic trees of the partial DPOL gene were constructed using the amino acid sequences of the detected herpesviruses with other alpha-, beta-, and gammaherpesviruses identified in bats worldwide (Figure 1). The Zambian bat betaherpesviruses were phylogenetically divided into seven groups (BHV-G1 to -G7) as shown in Figure 1B. BHV-G1, -G2, and -G7 each consisted of a single virus identified in *Macronycteris vittatus* bats. BHV-G3, -G4, and -G7 were phylogenetically related to bat betaherpesviruses previously identified in various countries. BHV-G3 formed a cluster near the branches of BHV-G2 and a betaherpesvirus detected in *Molossus molossus* bats in Uruguay. BHV-G4 formed a cluster with betaherpesviruses from *Rousettus aegyptiacus* bats in South Africa and Hungary, and BHV-G7 was closely related to betaherpesviruses in *Rhinolophus ferrumequinum* bats in China and Spain. BHV-G5 formed a unique lineage near the branch of murid betaherpesvirus 8. Of note, BHV-G6 contained betaherpesviruses identified in *Rousettus aegyptiacus* and *Macronycteris vittatus* bats and formed an independent cluster under the subfamily *Betaherpsvirinae*. The *Rousettus aegyptiacus*/Zambia/3/2018 virus shared 100% nucleotide identity with the *Macronycteris vittatus*/Zambia/114/2018 virus, suggesting that the viruses were distributed in both frugivorous and insectivorous bats. On the other hand, the Zambian bat gammaherpesviruses were phylogenetically divided into five groups (GHV-G1 to -G5) as shown in Figure 1C. GHV-G1, -G2, -G3, and -G4 each formed independent clusters or branches. GHV-G1 were monophyletic along with a gammaherpesvirus detected in *Rhinophus blythi* bats in China. GHV-G2 belonged to a large clade including GHV-G1 and gammaherpesviruses identified in various insectivorous bats in China and Japan. GHV-G3 and G4 formed distinct lineages within the *Betaherpsvirinae* clade. GHV-G5 were phylogenetically related to previously described gammaherpesviruses detected in frugivorous bats in Hungary and Bangladesh. These results indicated that genetically diverse herpesviruses were maintained in bats in the cave in Zambia.

### 3.3. Isolation of a Novel Gammaherpesvirus from Macronycteris Vittatus Bats

To isolate the viruses detected by genetic screening, the herpesvirus-positive swab samples were inoculated onto Vero E6 cells. Gammaherpesviruses were isolated from seven *Macronycteris vittatus* bats (sample IDs 59, 80, 82, 86, 88, 89, and 106). After several passages, a cytopathic effect (CPE) was observed in Vero E6 cells infected with the isolates (Figure 2). The isolated viruses were genetically analyzed by PCR and sequencing targeting the partial DPOL gene which were used for screening. According to the sequencing analyses, the seven isolates were divided into two closely related groups (i.e., sample IDs 59/80/86/89/106 and IDs 82/88). The two sequences showed 98.3% similarity (169/172 bp). These newly isolated herpesvirus strains were tentatively named Macronycteris gammaherpesvirus 1 (MaGHV1). The two representative MaGHV1 strains from sample IDs 80 and 82 (strains 80 and 82, respectively) were used for further analyses.

### 3.4. Complete Genome of MaGHV1 and Genetic Comparison between Strains

To determine the complete genome sequences of the MaGHV1 strains, we performed whole genome sequencing. The genome of MaGHV1 strain 82 was 116,972 bp in length with a GC content of 37.3% (Figure 3). The genome structure comprised a unique region that was flanked by tandem terminal repeats at each end (Table 3). The size of the consensus repeat sequences at both termini was 284 bp and the copy numbers were 7 and 3.8 at 5′ and 3′ termini, respectively. In the unique region, two repeat regions at coordinates 25,665 to 26,556 (repeat region 1) and 105,538 to 106,545 (repeat region 2) were found. A total of 79 ORFs were identified and most of the ORFs showed the highest identities to viral genes of rhinolophus gammaherpesvirus 1 (RGHV1) isolated from a greater horseshoe bat (*Rhinolophus ferrumequinum*) in Japan [20], followed by myotis gammaherpesvirus 8 isolated from a microbat (*Myotis velifer incautus*) cell line [18]. Notably, at least 67 ORFs showed homology with viral genes of equid herpesvirus 2, which is a well-known gammaherpesvirus, and one of them (ORF38) shared the highest identity (43.3%) with the myristylated tegument protein of equid gammaherpesvirus 2 rather than RGHV1. The BM1, BM4, and BM6 genes showed sequence similarity to cellular genes, CASP8 and FADD-like apoptosis regulator (32.5% identity), E3 ubiquitin-protein ligase (38.9% identity), and bcl-2 like protein (34.0% identity), respectively, and the BM4 and BM6 genes also had homologies to ORF 4 and ORF13 of RGHV1. Six ORFs (BM2, BM10, and BM13-16) showed no significant similarity to any known genes. In addition, ORF73, which encoded the homolog of nuclear antigen LANA-1, was mapped to a direct repeat (repeat region 2). To identify herpesviruses genetically related to MaGHV1, BLAST analyses were performed using the deduced amino acid sequences of viral proteins. The complete DPOL sequence (ORF9) of MaGHV1 strain 82 shared the highest identity (78.4%) with RGHV1. Among other ORFs, the deduced amino acid sequences of glycoprotein B (gB; ORF8), helicase-primase helicase subunit (ORF44), Uracil-DNA glycosylase (ORF46), deoxyribonuclease (ORF37), and major capsid protein (ORF25) showed 72.0, 78.6, 75.8, 68.0, and 78.1% identities to RGHV1 homologs, respectively (Appendix A). 

On the other hand, most of the 79 ORFs of MaGHV1 described above were also identified in the genome of strain 80: BM3–7 and ORF6–11 encoded in contig 1 (21,962 bp), BM8–13, ORF17–40, ORF42–50, and ORF52–70 in contig 2 (79,285 bp), and ORF73–75, BM14–16 in contig 3 (8979 bp) (Appendix A). Identity comparison of each viral protein between the MaGHV1 strains demonstrated > 90% amino acid identity of 70 ORFs. Relatively low identities (52.6–89.5%) were found in the remaining viral proteins, including three membrane proteins (BM3, ORF47, and BM11), one tegument protein (ORF45), and two hypothetical proteins (BM12 and BM13).

### 3.5. Phylogenetic Characterization of MaGHV1

Phylogenetic trees were constructed on the basis of DPOL and gB amino acid sequences. The phylogenetic analysis of DPOL demonstrated that two MaGHV1 strains were in the subfamily *Gammaherpesvirinae* and formed an independent lineage sharing a common origin with other bat-derived gammaherpesviruses such as RGHV1, myotis gammaherpesvirus 8, and myotis ricketti herpesvirus 2 detected by virome analyses in China [31] (Figure 4A). Similar results were obtained in the phylogenetic tree of gB (Figure 4B). In both the DPOL and gB phylogenetic trees, MaGHV1 strains were located near the branches of equid gammaherpesvirus 2 and equid gammaherpesvirus 5.

## 4. Discussion

In this study, we isolated a novel gammaherpesvirus, tentatively named MaGHV1, from *Macronycteris vittatus* bats in Zambia. We determined its complete genome sequence including the repeat sequences, and genetically characterized the encoded sequences. During the past two decades, many partial herpesvirus genes have been detected in various bats worldwide using NGS or PCR-based approaches [8,13,31,32,33]. However, few investigations isolated infectious herpesviruses. Isolated infectious viruses are often needed for downstream studies aiming at biological characterization of novel viruses, such as their host tropisms, transmissibility, and pathogenicity.

To date, three gammaherpesviruses (myotis gammaherpesvirus 8, eptesicus fuscus gammaherpesvirus, and RGHV1) have been isolated from *Myotis velifer incautus*, *Eptesicus fuscus*, and *Rhinolophus ferrumequinum* bats in the United States of America, Canada, and Japan, respectively [18,19,20]. Thus, MaGHV1 is the fourth infectious gammaherpesvirus strain isolated from bats. The deduced amino acid sequences of each MaGHV1 ORF showed less than 81% amino acid sequence identity to those of RGHV1, indicating that MaGHV1 is a distinct virus from other known herpesviruses. Phylogenetic analyses of full-length DPOL and gB amino acid sequences demonstrated that MaGHV1 was monophyletic with the other bat gammaherpesviruses except eptesicus fuscus gammaherpesvirus, and located near the branches of equid gammaherpesvirus 2 and equid gammaherpesvirus 5. The subfamily *Gammaherpesvirinae* is divided into seven genera: *Percavirus*, *Rhadinovirus* (e.g., Kaposi’s sarcoma-associated herpesvirus), *Lymphocryptovirus* (e.g., Epstein–Barr virus), *Macavirus* (e.g., ovine gammaherpesvirus 2), *Bossavirus* (e.g., delphinid gammaherpesvirus 1), *Manticavirus* (e.g., vombatid gammaherpesvirus 1), and *Patagivirus* (e.g., eptesicus fuscus gammaherpesvirus) [6]. To date, seven species have been classified into the genus *Percavirus*: *Equid gammaherpesvirus 2*, *Equid gammaherpesvirus 5*, *Phocid gammaherpesvirus 3* (harp seal herpesvirus), *Felid gammaherpesvirus 1* (felis catus gammaherpesvirus 1), *Mustelid gammaherpesvirus 1*, *Rhinolophid gammaherpesvirus 1* (RGHV1), and *Vespertilionid gammaherpesvirus 1* (myotis gammaherpesvirus 8). According to the species demarcation criteria of the ICTV [6], herpesviruses that have distinct epidemiological or biological characteristics and distinct genomes that represent independent replicating lineages differing from known species can be recognized as novel species. Therefore, we propose that MaGHV1 is a novel species belonging to the genus *Percavirus*.

Herpesviruses are highly disseminated in nature and most animal species have yielded at least one herpesvirus and frequently, several distinct herpesviruses [5]. By our PCR screening, at least seven betaherpesviruses and five gammaherpesviruses were detected in three Zambian bat species: *Rousettus aegyptiacus*, *Macronycteris vittatus*, and *Hipposideros caffer*. Our phylogenetic analyses indicate that diverse herpesviruses are distributed in the bat population in Zambia. It is generally believed that mammalian herpesviruses are mainly host-specific and have coevolved with their specific host species [34]. In a previous study, host–virus cophylogenetic analyses demonstrated that the phylogenetic tree of bat betaherpesviruses showed congruence with features in the phylogeny of corresponding host organisms, providing evidence for coevolutionary development of virus and host lineages [35]. Zambian bat herpesviruses were phylogenetically divided into several groups depending on their host species, except for BHV-G6. Interestingly, the nucleotide sequence of one betaherpesvirus from *Rousettus aegyptiacus* frugivorous bats was identical to that from *Macronycteris vittatus* insectivorous bats. Although evolutionary analyses of bat gammaherpesviruses have shown that host-switching events occur frequently [36], little information about the cross-species transmission of bat betaherpesviruses is available. In an epidemiological survey of herpesviruses in bats in Peru, betaherpesviruses detected in an *Artibeus lituratus* fruit bat and *Desmodus rotundus* vampire bats phylogenetically formed a cluster, suggesting a possible cross-species infection within the family *Phyllostomidae* (between the genera *Artibeus* and *Desmodus*) [35]. Our Zambian case may also show possible cross-species infection within order *Chiroptera* (between the families *Hipposideridae* and *Pteropodidae*), suggesting a unique and rare instance of cross-species transmission of betaherpesviruses. In our study, bats were captured using a harp trap, and trapped bats closely contacted each other in the collection chamber for a few hours. Considering this situation, bat saliva might be transferred to the oral cavities of other bats, implying possible contamination of the virus in an improper host without actual infection. To clarify the true frequency of betaherpesvirus transmission between the families *Hipposideridae* and *Pteropodidae* bat species, further investigation with virus isolation is required.

It is important to characterize herpesviruses in various host species for better understanding of the evolutionary history and host specificity of herpesviruses. Although partial herpesvirus genes have been detected in bats in various African countries [9,10,11,12,13], *Macronycteris vittatus* and *Hipposideros caffer* bats, which live in Africa, had not been analyzed for the detection of herpesviruses. In this study, we carried out the first genetic characterization of herpesviruses from these insectivorous bats living in Africa. Having a high Chiroptera (bat) diversity, there are currently 18 families (including 5 extinct), 84 genera (25 extinct), and 403 species (54 extinct) recognized as occurring in Africa [37]. Some of the bat species, including *Macronycteris vittatus* and *Hipposideros caffer*, were distributed only in Africa and the southwestern Arabian Peninsula, including Saudi Arabia and Yemen [37,38]. Therefore, surveillance of herpesviruses in bats in each country or region is necessary to fully understand the evolutionary host–virus relationship. 

The prototype of gammaherpesvirus, Kaposi’s sarcoma-associated herpesvirus, is estimated to encode over 85 genes, of which about 58 are conserved in all gammaherpesvirus genomes, including 40 core genes for all herpesviruses [6,7,39]. Overall, the structure of the MaGHV1 genome is similar to other gammaherpesviruses such as RGHV1, equid herpesvirus 2, and equid herpesvirus 5, and MaGHV1 carries all the conserved ORFs among the subfamily *Gammaherpesvirinae*. The conserved genes in the MaGHV1 genome are arranged colinearly in direction and position, mostly as in other gammaherpesvirus genomes. We found unique putative ORFs of MaGHV1, some of which were defined as hypothetical proteins with no homology with any known proteins, and it would be of interest to analyze the functions of these unique viral proteins in future studies. The MaGHV1 genome was flanked by the terminal repeat region consisting of multiple copies of tandemly repeated sequences. According to the ICTV criteria for the classification of the herpesvirus genome [6], MaGHV1 could be classified into the class 2 genome structure, which is common in members of the subfamily *Gammaherpesvirinae* [6]. The genome structure of RGHV1, which is genetically related to MaGHV1, could also be classified into the class 2 genome [20]. On the other hand, myotis gammaherpesvirus 8, which might share a common origin with MaGHV1, is currently unclassified because its terminal repeat sequences were not determined. Therefore, complete genome sequences of herpesviruses, including the repeat regions, are necessary to clarify the genetic relationships among diverse herpesviruses.

In the present study, we showed the presence of multiple bat-derived herpesviruses in Zambia. Our findings provide new information regarding the genetic diversity of bat herpesviruses. Similar prevalence patterns were also observed in bats in other countries such as Indonesia and South Africa [11,32]. As the first investigation on bat-derived herpesviruses in Zambia, we only screened three bat species in the present study. However, since there are 65 known bat species in Zambia [40], more diverse herpesviruses might be associated with various bat species in the same manner as other bat herpesviruses that have been previously recognized. It would be of interest to further investigate the prevalence of herpesvirus infection of bats in Zambia and other African countries.

## Figures and Tables

**Figure 1 viruses-15-01369-f001:**
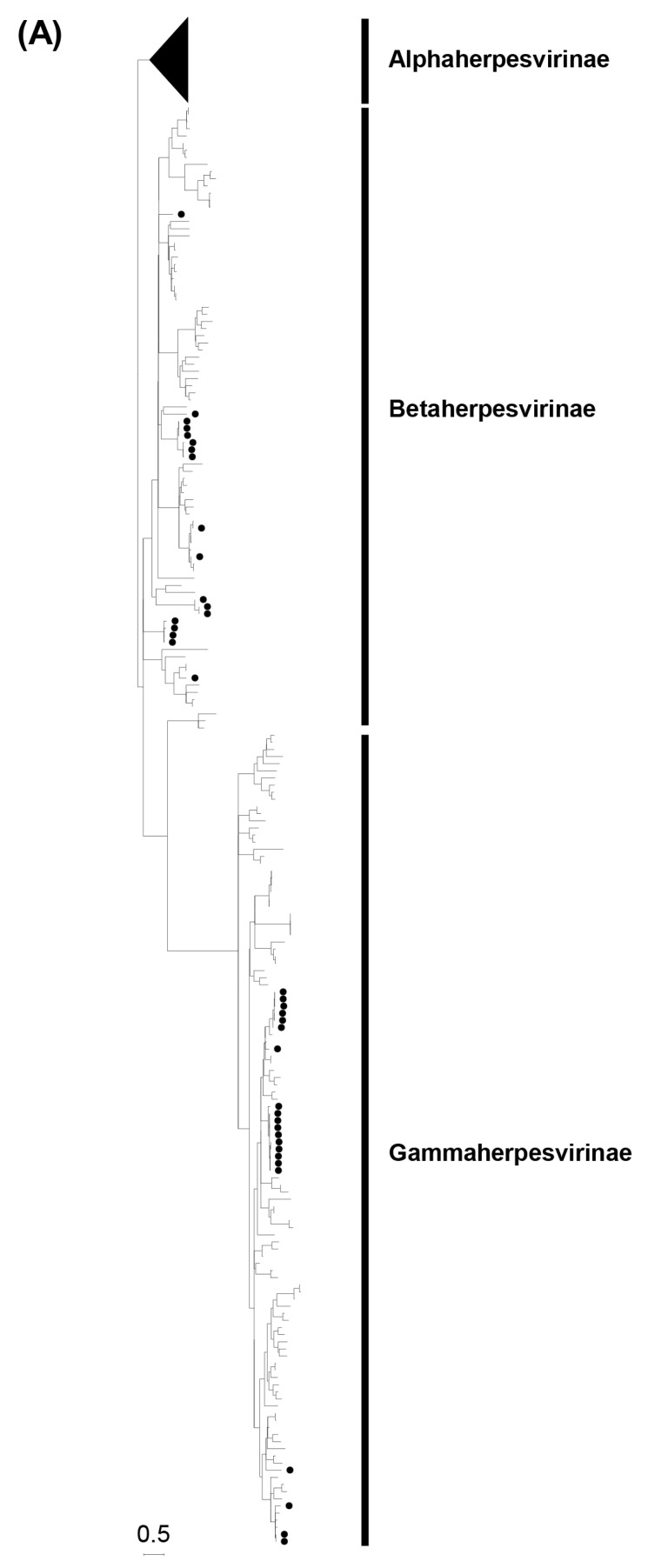
Phylogenetic analysis of the partial DPOL gene sequences obtained by screening. A phylogenetic tree based on the predicted amino acid sequences of the DPOL corresponding to positions 690–746 of the equid gammaherpesvirus 2 DNA polymerase catalytic subunit (GenBank accession no. NC_001650.2) was constructed using the maximum likelihood method with 1000 bootstrap replicates (**A**). Betaherpesviruses and gammaherpesviruses are magnified in panels (**B**) and (**C**), respectively. Bootstrap values greater than 50% are shown on the interior branch nodes and scale bars indicate the number of substitutions per site. Betaherpesvirus (BHV) and gammaherpesvirus (GHV) groups are indicated. Individual herpesviruses detected in bats are highlighted and named as follows: accession number/bat species of origin/country of identification/common name/year of collection. The black dots represent the bat herpesviruses detected in this study.

**Figure 2 viruses-15-01369-f002:**
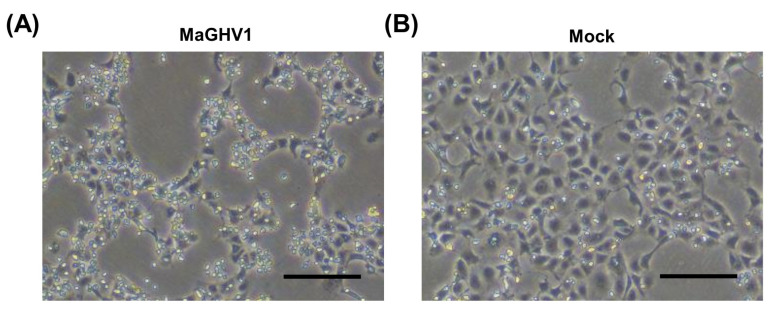
Isolation of MaGHV1. Phase-contrast images of MaGHV1 strain 82-infected (**A**) and mock-infected Vero E6 cells (**B**) at 12 days post infection. The cells infected with the virus exhibited cytopathic effects. Scale bar, 200 µm.

**Figure 3 viruses-15-01369-f003:**
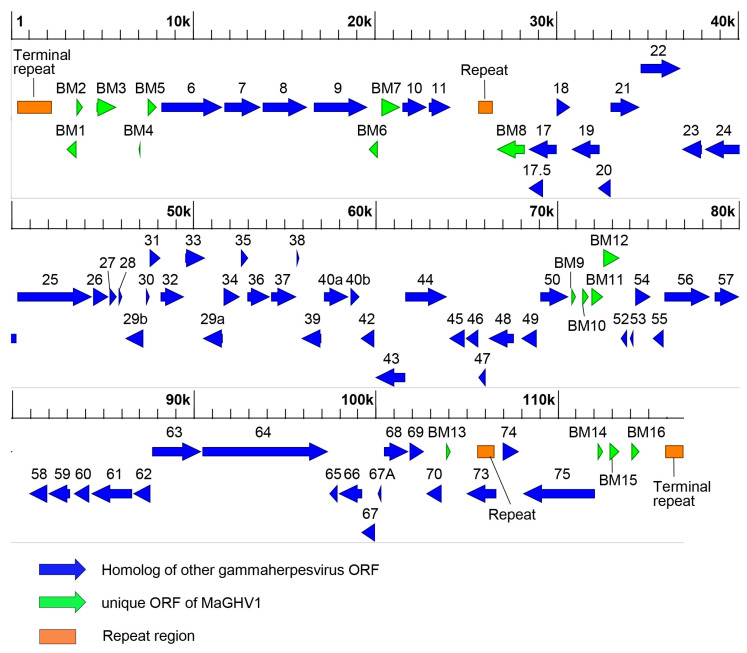
Map of the MaGHV1 genome. Predicted ORFs in the genome of MaGHV1 strain 82 are indicated by arrows. Blue arrows represent homologs of other gammaherpesvirus ORFs. Green arrows represent unique ORFs of MaGHV1: ORF with homology to cellular genes and/or viral protein genes encoding in some gammaherpesviruses; ORF with no homology to known proteins. Repeat regions found by the tandem repeat finder program with the copy number > 2.0 and the score > 250 are indicated by orange rectangles. The total length of the genome is 116,972 bp. The scale is shown at the top.

**Figure 4 viruses-15-01369-f004:**
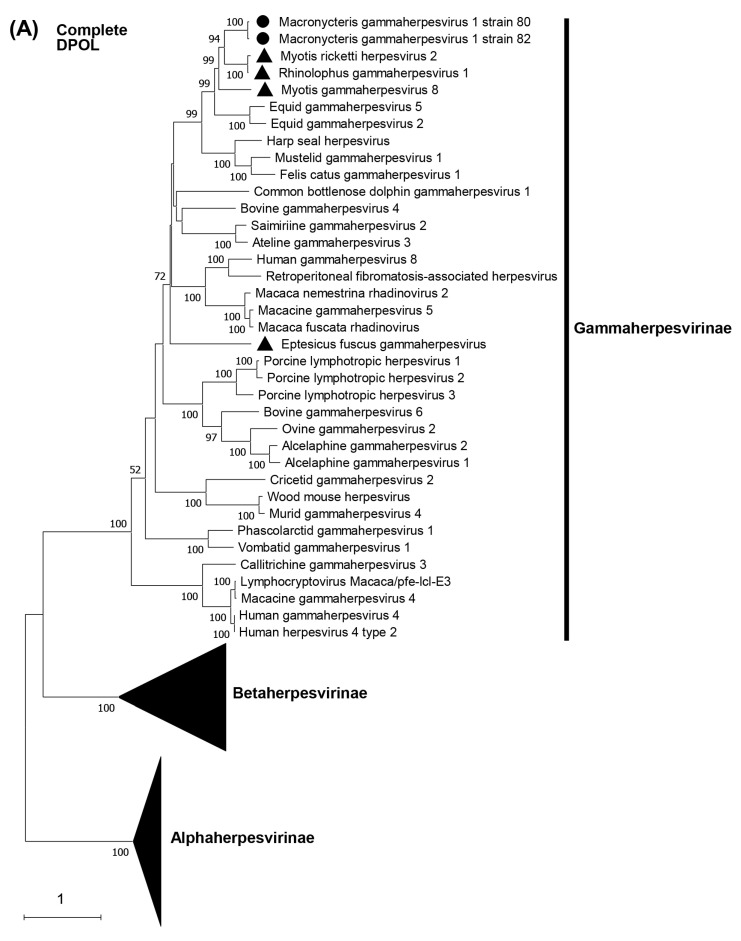
Phylogenetic analysis of the complete DPOL and gB. The complete ORF amino acid sequences of DPOL and gB of MaGHV1 were phylogenetically analyzed with corresponding proteins from the representative alpha-, beta-, and gammaherpesviruses. Bootstrap values greater than 50% are shown on the interior branch nodes and scale bars indicate the number of substitutions per site. The black dots represent MaGHV1 isolated in this study. The triangles represent herpesviruses identified in bats.

**Table 1 viruses-15-01369-t001:** Summary of bat samplings in 2018 in Zambia and screening results of herpesviruses.

Sampling Date	Bat (Species)	Sample ID	No. of Positive/No. of Tested Samples (%)
19 November	Egyptian fruit bat (*Rousettus aegyptiacus*)	1–17	7/17 (41.2)
	Striped leaf-nosed bat (*Macronycteris vittatus*)	18–70	39/53 (73.6)
22 November	Egyptian fruit bat (*Rousettus aegyptiacus*)	71–77	0/7 (0)
	Striped leaf-nosed bat (*Macronycteris vittatus*)	78–111 113–130	43/52 (82.7)
	Sundevall’s roundleaf bat (*Hipposideros caffer*)	112	1/1 (100)
Total			90/130 (69.2)

**Table 2 viruses-15-01369-t002:** Summary of herpesvirus DPOL-positive bats detected by PCR screening.

Virus	Accession No.	Sample ID	Subfamily	Group
*Macronycteris vittatus*/Zambia/64/2018	LC762209	64	*Betaherpesvirinae*	BHV-G1
*Macronycteris vittatus*/Zambia/50/2018	LC762210	50, 69	*Betaherpesvirinae*	BHV-G2
*Macronycteris vittatus*/Zambia/20/2018	LC762211	20	*Betaherpesvirinae*	BHV-G3
*Macronycteris vittatus*/Zambia/36/2018	LC762212	36	*Betaherpesvirinae*	BHV-G3
*Macronycteris vittatus*/Zambia/41/2018	LC762213	41	*Betaherpesvirinae*	BHV-G3
*Macronycteris vittatus*/Zambia/46/2018	LC762214	46	*Betaherpesvirinae*	BHV-G3
*Macronycteris vittatus*/Zambia/101/2018	LC762215	101, 111	*Betaherpesvirinae*	BHV-G3
*Macronycteris vittatus*/Zambia/104/2018	LC762216	104	*Betaherpesvirinae*	BHV-G3
*Rousettus aegyptiacus*/Zambia/10/2018	LC762217	10	*Betaherpesvirinae*	BHV-G4
*Rousettus aegyptiacus*/Zambia/11/2018	LC762218	11	*Betaherpesvirinae*	BHV-G4
*Macronycteris vittatus*/Zambia/37/2018	LC762219	37, 44	*Betaherpesvirinae*	BHV-G5
*Macronycteris vittatus*/Zambia/49/2018	LC762220	49, 56, 61, 66, 89, 103, 121, 123, 128	*Betaherpesvirinae*	BHV-G5
*Macronycteris vittatus*/Zambia/106/2018	LC762221	106, 109	*Betaherpesvirinae*	BHV-G5
*Rousettus aegyptiacus*/Zambia/3/2018	LC762222	3	*Betaherpesvirinae*	BHV-G6
*Rousettus aegyptiacus*/Zambia/6/2018	LC762223	6	*Betaherpesvirinae*	BHV-G6
*Macronycteris vittatus*/Zambia/114/2018	LC762224	114	*Betaherpesvirinae*	BHV-G6
*Macronycteris vittatus*/Zambia/126/2018	LC762225	126	*Betaherpesvirinae*	BHV-G6
*Macronycteris vittatus*/Zambia/84/2018	LC762226	84	*Betaherpesvirinae*	BHV-G7
*Macronycteris vittatus*/Zambia/22/2018	LC762227	22, 58, 65, 81, 83, 110, 130	*Gammaherpesvirinae*	GHV-G1
*Macronycteris vittatus*/Zambia/30/2018	LC762228	30, 60	*Gammaherpesvirinae*	GHV-G1
*Macronycteris vittatus*/Zambia/38/2018	LC762229	38	*Gammaherpesvirinae*	GHV-G1
*Macronycteris vittatus*/Zambia/39/2018	LC762230	39	*Gammaherpesvirinae*	GHV-G1
*Macronycteris vittatus*/Zambia/80/2018	LC762231	59, 70, 80, 86, 91, 97, 98, 107, 113	*Gammaherpesvirinae*	GHV-G1
*Macronycteris vittatus*/Zambia/82/2018	LC762232	18, 24, 26, 31, 78, 82, 85, 88, 124, 129	*Gammaherpesvirinae*	GHV-G1
*Hipposideros caffer*/Zambia/112/2018	LC762233	112	*Gammaherpesvirinae*	GHV-G2
*Macronycteris vittatus*/Zambia/21/2018	LC762234	21	*Gammaherpesvirinae*	GHV-G3
*Macronycteris vittatus*/Zambia/23/2018	LC762235	23, 43, 118	*Gammaherpesvirinae*	GHV-G3
*Macronycteris vittatus*/Zambia/28/2018	LC762236	28, 99, 100, 116, 120	*Gammaherpesvirinae*	GHV-G3
*Macronycteris vittatus*/Zambia/29/2018	LC762237	29	*Gammaherpesvirinae*	GHV-G3
*Macronycteris vittatus*/Zambia/42/2018	LC762238	42	*Gammaherpesvirinae*	GHV-G3
*Macronycteris vittatus*/Zambia/45/2018	LC762239	45	*Gammaherpesvirinae*	GHV-G3
*Macronycteris vittatus*/Zambia/55/2018	LC762240	55, 119	*Gammaherpesvirinae*	GHV-G3
*Macronycteris vittatus*/Zambia/92/2018	LC762241	92	*Gammaherpesvirinae*	GHV-G3
*Macronycteris vittatus*/Zambia/94/2018	LC762242	94	*Gammaherpesvirinae*	GHV-G3
*Macronycteris vittatus*/Zambia/96/2018	LC762243	96	*Gammaherpesvirinae*	GHV-G3
*Macronycteris vittatus*/Zambia/25/2018	LC762244	25, 33, 34, 47, 52, 95, 105, 108, 125	*Gammaherpesvirinae*	GHV-G4
*Rousettus aegyptiacus*/Zambia/4/2018	LC762245	4	*Gammaherpesvirinae*	GHV-G5
*Rousettus aegyptiacus*/Zambia/7/2018	LC762246	7	*Gammaherpesvirinae*	GHV-G5
*Rousettus aegyptiacus*/Zambia/15/2018	LC762247	15	*Gammaherpesvirinae*	GHV-G5

Abbreviations: BHV, betaherpesvirus; GHV, gammaherpesvirus.

**Table 3 viruses-15-01369-t003:** Predicted protein coding regions and repeat regions in the genome of Macronycteris gammaherpesvirus 1 strain 82.

Gene	Location (Nucleotides)	Strand	Size (No. of Amino Acids)	Product or Predicted Function
BM1	3031–3630	−	199	homolog of CASP8 and FADD-like apoptosis regulator
BM2	3577–3990	+	137	hypothetical protein
BM3	4693–5832	+	379	homolog of EHV2 E3 membrane protein E3
BM4	6988–7152	−	54	homolog of E3 ubiquitin-protein ligase and RGHV1 ORF4 MIR-like membrane protein
BM5	7520–8077	+	185	homolog of EHV2 E4 apoptosis regulator BALF1
ORF6	8279–11,677	+	1132	single-stranded DNA binding protein
ORF7	11,748–13,799	+	683	DNA packaging terminase subunit 2
ORF8	13,816–16,362	+	848	glycoprotein B
ORF9	16,633–19,638	+	1001	DNA polymerase catalytic subunit
BM6	19,673–20,185	−	170	bcl-2-like protein
BM7	20,383–21,486	+	367	homolog of EHV2 E6 membrane protein BILF1
ORF10	21,530–22,927	+	465	homolog of EHV2 ORF10 protein G10
ORF11	22,977–24,209	+	410	homolog of EHV2 ORF11 virion protein G11
BM8	26,676–28,304	−	542	homolog of bovine herpesvirus 6 Bov8 putative major envelope glycoprotein, RGHV1 ORF18 glycoprotein
ORF17	28,449–30,029	−	526	capsid maturation protease
ORF17.5	28,449–29,300	−	284	capsid scaffold protein
ORF18	30,022–30,798	+	258	homolog of EHV2 ORF18 protein UL79
ORF19	30,795–32,417	−	540	homolog of EHV2 ORF19 DNA packaging tegument protein UL25
ORF20	32,281–32,985	−	234	homolog of EHV2 ORF20 nuclear protein UL24
ORF21	32,984–34,600	+	538	thymidine kinase
ORF22	34,600–36,861	+	753	glycoprotein H
ORF23	36,858–38,051	−	397	homolog of EHV2 ORF23 tegument protein UL88
ORF24	38,106–40,298	−	730	homolog of EHV2 ORF24 protein UL87
ORF25	40,303–44,436	+	1377	major capsid protein
ORF26	44,456–45,358	+	300	capsid triplex subunit 2
ORF27	45,359–45,802	+	147	homolog of EHV2 ORF27 envelope glycoprotein 48
ORF28	45,867–46,118	+	83	homolog of EHV2 ORF28 envelope glycoprotein 150
ORF29a	50,460–51,623	−	387	DNA packaging terminase subunit 1
ORF29b	46,207–47,250	−	347	DNA packaging terminase subunit 1
ORF30	47,385–47,639	+	84	homolog of EHV2 ORF30 protein UL91
ORF31	47,552–48,235	+	227	homolog of EHV2 ORF31 protein UL92
ORF32	48,181–49,536	+	451	homolog of EHV2 ORF32 DNA packaging tegument protein UL17
ORF33	49,529–50,695	+	388	homolog of EHV2 ORF33 tegument protein UL16
ORF34	51,622–52,590	+	322	homolog of EHV2 ORF34 protein UL95
ORF35	52,577–53,047	+	156	homolog of EHV2 ORF35 tegument protein UL14
ORF36	52,944–54,245	+	433	tegument serine/threonine protein kinase
ORF37	54,248–55,702	+	484	deoxyribonuclease
ORF38	55,657–55,854	+	65	myristylated tegument protein
ORF39	55,918–57,051	−	377	glycoprotein M
ORF40a	57,162–58,538	+	458	helicase-primase subunit
ORF40b	58,624–59,154	+	176	helicase-primase subunit
ORF42	59,149–59,964	−	271	homolog of EHV2 ORF42 tegument protein UL7
ORF43	59,951–61,654	−	567	capsid portal protein
ORF44	61,620–63,986	+	788	helicase-primase helicase subunit
ORF45	64,035–64,934	−	299	homolog of EHV2 ORF45 tegument protein G45
ORF46	64,936–65,694	−	252	uracil-DNA glycosylase
ORF47	65,657–66,115	−	152	glycoprotein L
ORF48	66,213–67,646	−	477	homolog of EHV2 ORF48 tegument protein G48
ORF49	68,010–68,894	−	294	homolog of EHV2 ORF49 tegument protein G49
ORF50	69,039–70,646	+	535	protein Rta
BM9	70,780–71,067	+	95	hypothetical protein, homolog of RGHV1 ORF53
BM10	71,349–71,759	+	136	hypothetical protein
BM11	71,849–72,568	+	239	homolog of EHV2 E7A envelope glycoprotein 42
BM12	72,526–73,479	+	317	hypothetical protein, homolog of RGHV1 ORF57
ORF52	73,493–73,897	−	134	homolog of EHV2 ORF52 virion protein G52
ORF53	73,952–74,212	−	86	glycoprotein N
ORF54	74,302–75,171	+	289	dUTPase
ORF55	75,240–75,893	−	217	homolog of EHV2 ORF55 tegument protein UL51
ORF56	75,866–78,433	+	855	helicase-primase primase subunit
ORF57	78,642–80,066	+	474	multifunctional expression regulator
ORF58	80,927–81,964	−	345	homolog of EHV2 ORF58 envelope protein UL43
ORF59	81,973–83,202	−	409	DNA polymerase processivity subunit
ORF60	83,364–84,281	−	305	ribonucleotide reductase subunit 2
ORF61	84,309–86,621	−	770	ribonucleotide reductase subunit 1
ORF62	86,645–87,646	−	333	capsid triplex subunit 1
ORF63	87,657–90,434	+	925	homolog of EHV2 ORF63 tegument protein UL37
ORF64	90,450–97,421	+	2323	large tegument protein
ORF65	97,435–97,935	−	166	small capsid protein
ORF66	97,946–99,256	−	436	homolog of EHV2 ORF66 protein UL49
ORF67	99,163–100,002	−	279	nuclear egress membrane protein
ORF67A	100,065–100,334	−	89	homolog of EHV2 ORF67A DNA packaging protein UL33
ORF68	100,437–101,819	+	460	envelope glycoprotein, DNA packaging protein UL32
ORF69	101,821–102,684	+	287	nuclear egress lamina protein
ORF70	102,750–103,628	−	292	thymidylate synthase
BM13	103,853–104,161	+	102	hypothetical protein
ORF73	104,958–106,661	−	567	nuclear antigen LANA-1
ORF74	106,940–107,914	+	324	homolog of EHV2 ORF74 membrane protein G74
ORF75	108,078–112,091	−	1337	homolog of EHV2 ORF75 tegument protein G75
BM14	112,188–112,517	+	109	hypothetical protein
BM15	112,844–113,437	+	197	hypothetical protein
BM16	114,035–114,538	+	167	hypothetical protein
Terminal repeat	319–2295	none	none	none
Repeat region 1	25,665–26,556	none	none	none
Repeat region 2	105,538–106,545	none	none	none
Terminal repeat	115,903–116,972	none	none	none

## Data Availability

All relevant data are provided in the manuscript.

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
