# Peer review of "Surveillance, Isolation, and Genetic Characterization of Bat Herpesviruses in Zambia"

_viruses, 2023, doi:10.3390/v15061369_

Round 1

Reviewer 1 Report

The manuscript “Surveillance, isolation, and genetic characterization of bat herpesviruses in Zambia” describes the presence of different beta- and gammaherpesviruses in several bat species of Zambia. The authors were able to isolate and obtain the complete genome sequence of a novel gammaherpesvirus. The methodology is appropriate, and the results are of interest for those interested on infectious agents in wildlife. I have only some minor comments:

Abstract

- Please provide the common name of the bat species when first mentioned in the Abstract and in the body of the manuscript.

- I recommend to avoid the use of “herpesvirus genomes” when you refer to the detection of a short fragment of DNA polymerase gene. You are not detecting the complete genome, only a small region. I suggest to replace it by “herpesvirus detection” or “herpesviral DNA polymerase amplification” along the text.

100% (1/1) of Hipossideros caffer….. It is not correct to wite 100% when you only tested one sample. Please modify to “and an Hipposideros caffer bat” and do the same in the body of the text. Please add the common name of the species.

Introduction

Please replace (the order Chioptera) by (order Chiroptera)

Recently, novel viral sequences… I suggest to start the sentence with “Novel viral sequences…”

Please provide a reference to the sentence: “Recently, novel viral sequences are constantly discovered in various bat species owing to the development of next-generation sequencing technologies (NGS).”

Results

“… we tentatively classified them into the subfamily the Betaherpesvirinae”, please replace by ““… we tentatively classified them into the subfamily Betaherpesvirinae”,

Table 2. Delete the word “genomes”. I suggest a sentence like “Summary of herpesvirus DPOL positive bat species detected PCR”. Delete the word “RT”. Please explain the meaning of the abbreviations BHV and GHV, as tables always should stand alone.  

Discussion

Provide only scientific name: striped leaf-nosed bats (Macronycteris vittatus)

81% sequence identity... Specify if you referred to nucleotide or amino acid identity

“In an epidemiological survey of herpesviruses in bats in Peru, betaherpesviruses...” Please correct to “betaherpesviruses”

The English requires some minor corrections. Please see the comment and suggestions fo Authors.

Author Response

Abstract

Point 1: Please provide the common name of the bat species when first mentioned in the Abstract and in the body of the manuscript.

Response 1: Following the reviewer’s comment, we added common names of the bat species on lines 49-52 and 116-117 in the revised manuscript.

Point 2: I recommend to avoid the use of “herpesvirus genomes” when you refer to the detection of a short fragment of DNA polymerase gene. You are not detecting the complete genome, only a small region. I suggest to replace it by “herpesvirus detection” or “herpesviral DNA polymerase amplification” along the text.

Response 2: Following the reviewer’s suggestion, we modified the sentences regarding the detected partial genomes (lines 50, 53, 101, 134, 135, 153, 215, 272, 353, 411, 412, and 453).

Point 3: 100% (1/1) of Hipossideros caffer….. It is not correct to wite 100% when you only tested one sample. Please modify to “and an Hipposideros caffer bat” and do the same in the body of the text. Please add the common name of the species.

Response 3: Following the reviewer’s comment, we modified the sentences (lines 51 and 217).

Introduction

Point 4: Please replace (the order Chioptera) by (order Chiroptera)

Response 4: According to the reviewer’s suggestion, we replace the term “the order Chiroptera” with “order Chiroptera” (lines 68 and 402).

Point 5: Recently, novel viral sequences… I suggest to start the sentence with “Novel viral sequences…”

Response 5: According to the reviewer’s suggestion, we deleted the word (line 70).

Point 6: Please provide a reference to the sentence: “Recently, novel viral sequences are constantly discovered in various bat species owing to the development of next-generation sequencing technologies (NGS).”

Response 6: Following the reviewer’s comment, we added the reference [1] to this sentence (line 72).

Results

Point 7: “… we tentatively classified them into the subfamily the Betaherpesvirinae”, please replace by ““… we tentatively classified them into the subfamily Betaherpesvirinae”,

Response 7: We corrected this typo (line 226).

Point 8: Table 2. Delete the word “genomes”. I suggest a sentence like “Summary of herpesvirus DPOL positive bat species detected PCR”. Delete the word “RT”. Please explain the meaning of the abbreviations BHV and GHV, as tables always should stand alone.

Response 8: We appreciate the reviewer’s suggestion. We added the explanation of these abbreviations and modified the caption of Table 2 as follow:

 “Table 2. Summary of herpesvirus DPOL-positive bats detected by PCR screening.”

Discussion

Point 9: Provide only scientific name: striped leaf-nosed bats (Macronycteris vittatus)

Response 9: According to the reviewer’s comment, we replaced the term “striped leaf-nosed bats (Macronycteris vittatus)” with “Macronycteris vittatus bats. (line 352)

Point 10: 81% sequence identity... Specify if you referred to nucleotide or amino acid identity

Response 10: Following the reviewer’s comment, we modified the term to “81% amino acid sequence identity”.  (line 364).

Point 11: “In an epidemiological survey of herpesviruses in bats in Peru, betaherpsviruses...” Please correct to “betaherpesviruses”

Response 11: Sorry for the typo. We modified the misspelling on line 399 in the revised manuscript.

Comments on the Quality of English Language

Point 12: The English requires some minor corrections. Please see the comment and suggestions fo Authors.

Response 12: We corrected and addressed all issues recognized by reviewers in the revised manuscript.

Reviewer 2 Report

Harima et al. presented a study detecting and isolating diverse herpesviruses in various bats in Zambia. This is a well-conducted study with clearly presented results. The data adds a lot to the field of bat virus investigation. The manuscript is well written, and the data generally supports the conclusions. I only have three suggestions.

In abstract, I don’t think the 100% on line 51 has any meaning given that only one sample was tested. I would revise this sentence to 1 Hipposideros caffer... So did line 218.

Figure 2 is not so clear. For me, it is difficult to tell the difference between infected and mock cells.

It would be great if the authors could provide a TEM picture of the isolated virus.

Author Response

Point 1: In abstract, I don’t think the 100% on line 51 has any meaning given that only one sample was tested. I would revise this sentence to 1 Hipposideros caffer... So did line 218.

Response 1: Following the reviewer’s comment, we modified these sentence in the revised manuscript (lines 51 and 217).

Point 2: Figure 2 is not so clear. For me, it is difficult to tell the difference between infected and mock cells.

Response 2: We appreciate the reviewer’s comment. According to the comment, we replaced the original images with the magnified images of MaGHV1 strain 82-infected and mock-infected Vero E6 cells in Figure 2.

Point 3: It would be great if the authors could provide a TEM picture of the isolated virus.

Response 3: Thank you for the reviewer’s suggestion. We isolated and stored the novel gammaherpesvirus at our laboratory in Zambia. Unfortunately, we could not perform a TEM analysis of the virus because no transmission electron microscope (TEM) is installed at the Zambian laboratory.
